# Factors affecting onion production: An empirical study in the Raya Kobo district, Amhara regional state, Ethiopia

**Mesele Belay Zegeye** [ID] * [◉], **Tesfahun Ayanaw Alemu** [◉], **Moges Asmare Sisay** [ID] [◉], **Sisay Genanu Mulaw** [◉], **Tadesse Wudu Abate** [◉]

Department of Economics, Woldia University, Woldia, Ethiopia

◉ These authors contributed equally to this work.
* mesele99belay@gmail.com

**Data Availability Statement:** The data that support the findings of this study is openly available in the link https://zenodo.org/uploads/11082435.

## Abstract

Onion is a vital vegetable crop in Ethiopia, with significant economic and health benefits. However, its production trend is not consistent, with periods of increase and decrease; and its productivity in the country falls far below its potential. As a result, farmers are not yet fully benefited from onion production. Thus, this study was initiated to identify the factors influencing onion production in the Raya Kobo District of Amhara Regional State of Ethiopia. Data was collected from 189 onion-producing farmers through household surveys, and both descriptive and econometric techniques were used for analysis. The study found significant variation in onion production among farmers, with lower levels compared to national and international averages. Factors such as gender, education level, experience, labor force, land size, access to extension services, irrigation water, land plough frequency, and fertilizer availability positively impact onion production. However, excessive fertilizer use was found to have a negative effect. The study also identified challenges faced by farmers, including input shortages, high costs, diseases, labor issues, soil infertility, and storage knowledge gaps. The study recommends policymakers and stakeholders to utilize these findings to develop effective policies and interventions that can enhance onion production, benefiting farmers and improving the overall onion production.

## 1. Introduction

Since agriculture is the primary means of subsistence for the majority of the poor in developing nations, particularly those in Sub-Saharan Africa, it is widely recognized as a pivotal economic sector for fostering economic growth, eradicating poverty, and enhancing food security [1, 2]. Ethiopia, situated in Sub-Saharan Africa, heavily relies on rainfed agriculture, which contributes 34.1% to its Gross Domestic Product, engages 79% of the labor force, generates 79% of foreign exchange earnings and serves as a critical source of capital and raw materials for investment in other sectors and markets [2–4]. Smallholder agriculture covers 95% of the cropped area and produces over 90% of agricultural output. However, Ethiopian agriculture is

**Funding:** The author(s) received no specific funding for this work.

**Competing interests:** The authors have declared that no competing interests exist.

vulnerable to rainfall fluctuations and is experiencing increasing land fragmentation [5]. The sector's contribution to the economy is declining due to reliance on traditional farming practices, limited use of modern technologies, and low productivity [4, 6, 7]. Consequently, escalating food demand and prices have led to food shortages, insecurity, and widespread poverty. Addressing poverty and improving food security are critical concerns, requiring a focus on enhancing agricultural production and productivity [7, 8].

Onion is a significant cash crop in Ethiopia, particularly in the Amhara Region. It is a crucial source of income, contributes to foreign exchange earnings, and improves the livelihoods of smallholder farmers. Onion is also essential for enhancing flavors in Ethiopian cuisine, particularly in daily stews known as "*Wot*" and vegetable-based meals [9]. Additionally, it provides health benefits by neutralizing acidic substances during digestion [10, 11]. Ethiopia is the largest onion producer in Sub-Saharan Africa, with vast potential for onion production. Around 36.4 million hectares of land are allocated to onion cultivation, resulting in a harvest of approximately 346,048.1 tons of bulbs, with a productivity rate of 8.9 tons per hectare. Most of the onion production (73%) is consumed by farm households, while 26% is sold in the market and 1% is used for seed purposes [12].

However, Ethiopian farmers face challenges in achieving production levels comparable to other countries. The average onion production intensity in Ethiopia is 8.9 tons per hectare, lower than global averages of 19.1 tons per hectare; and 35.5 tons per hectare in Egypt, and 18 tons per hectare in Sudan [12, 13]. Within Ethiopia, the Oromia Region leads in onion production with a rate of 19.32 tons per hectare, followed by the Amhara Region. The Amhara Region contributes around 50% of the national onion production, with a productivity rate of 12.3 tons per hectare. The Central Gondar, East Gojjam, North Wollo, South Wollo, and South Gondar Zones within the Amhara Region are major onion production areas [14, 15].

The data clearly show that onion production and productivity in Ethiopia, especially in the Amhara region, are low and insufficient. This is because that the Amhara region faces significant agricultural challenges, including the absence of improved seed varieties, limited access to quality seeds and production technologies, inadequate irrigation services, lack of tools and fertilizers, poor storage facilities, poor handling practices, hidden price information, lack of coordination, poor market linkages, and poor quality standards. In contrast, the neighboring Oromia region appears to have better access to agricultural inputs, infrastructure, and market connections, enabling its farmers to achieve greater onion productivity [7, 24]. Farmer inexperience, illiteracy, and related factors also contribute to inefficiency [15–17]. There is also a significant increase in onion demand in Ethiopia, resulting in a supply-demand mismatch [18].

Several studies conducted at global, national, and regional levels have investigated factors influencing onion production, productivity, and technical efficiency. These studies include [11, 14, 15, 17, 19–28] and [29]. The studies mentioned above have found that factors such as gender, age, education level, household size, onion purchasing quantity, production experience, land area allocated to onion cultivation, extension services, storage facilities, coordination among producers, access to fertilizers and quality seeds, irrigation services, diseases and pests, plant spacing, soil fertility management practices, and access to credit significantly influence onion production.

This paper contributes to the existing literature in four main ways. Firstly, while previous studies (e.g.[11, 14, 15, 22, 28] and [19, 24]), have focused on technical efficiency, economic feasibility, profitability, onion marketing, and the influence of seedling age and variety on onion growth, there is a need for additional research considering the constraints of onion production in Ethiopia, particularly in the Amhara region. Secondly, existing studies, such as [29], have only used descriptive statistics to examine the rise and fall of onion production in Northwest Ethiopia, making it difficult to identify the significant factors influencing onion

production. Thirdly, this study introduces additional variables, namely the frequency of onion land ploughing, sowing fertilizer, and hoe usage, which have a significant effect on onion production but were not considered in previous studies. Lastly, although the Raya Kobo district is recognized as a major onion producer, no specific investigation into the determinants of onion production has been conducted or documented in the study area. Therefore, this study aims to fill this research gap by investigating the determinants of onion production in the Raya Kobo district of the Amhara regional state in Ethiopia.

## 2. Methodology

### 2.1. Study area profile

Raya Kobo is a District located in the northeastern part of the North Wollo Zone, within the Amhara Region of Ethiopia. It shares its borders with the Habru and Guba Lafto Districts to the south, the Gidan district to the west, the Tigray region to the north, and the Afar region to the east. Geographically, it falls between the latitudes of 11˚ 54' 04" and 12˚ 20' 56" N, and longitudes of 39˚ 25' 56" and 39˚ 49' 04" E. The district has an altitude that ranges from 1400 to 3100 m above sea level, as presented in Fig 1. According to [30], the total cultivated land area in the Raya Kobo District is 58,045 hectares, with 90% of it relying on rainfall for irrigation. On average, each land holding in the District measures 1.25 hectares, while the average area under irrigation is 0.15 hectares. During the rainy season, the main crops cultivated in the District include Teff, Sorghum, Maize, and Chickpea, with their respective areas of cultivation. However, due to the dependency on rainfall, it is challenging to grow vegetables and staple crops under rain-fed conditions unless supplemented with irrigation water. Consequently, the production of these crops is limited to households with access to irrigation facilities. In the District's irrigable lands, horticultural crops, particularly onions, play a vital role in ensuring food security for households. Alongside onions, tomatoes and peppers are also cultivated

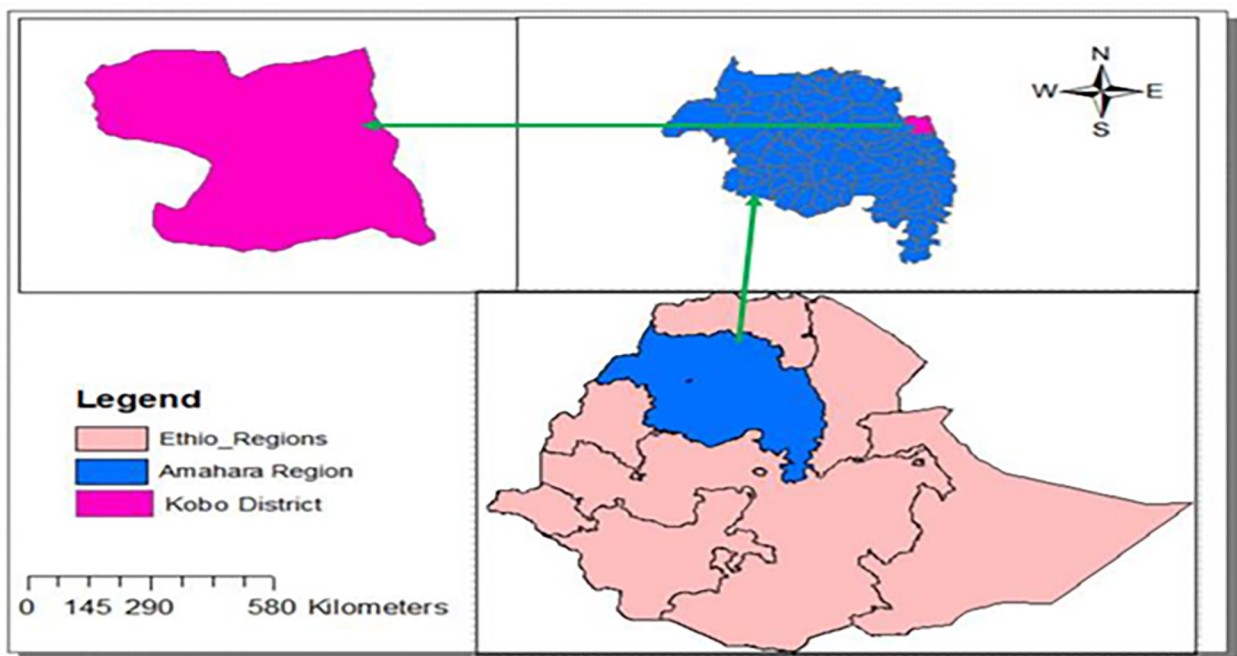

**Fig 1. Map of the study area.**

using irrigation methods. However, onions hold the largest share of the cultivated area. In fact, more than 75% of the irrigated land in the District is allocated to onion production, providing a source of income for vulnerable households [19]. Besides, According to [30] in 2017/18 production season total production of red onion in the district was 428,700 quintals on 2,882 hectares of land. Furthermore, in 2020, the average annual onion production in the District was 68.127 quintals per hectare. Considering the significance of Raya Kobo as a potential onion-producing District in the North Wollo Zone, this study focuses on five major onion producing kebeles: Adiskigni, Jarota, Kobo Zuria, Aradum, and Abuare [19, 30].

## 2.2. Data types, sources, and collection methods

The study uses primary data collected using a structured questionnaire from onion producing farm households in 2023. A structured questionnaire was designed and distributed to farm households in the study area to obtain information on demographic and socio-economic characteristics onion producers, onion production practices, access to production inputs and the challenges faced by farmers to produce onion in the study area. To ensure a comprehensive understanding of the respondents' perspectives, the questionnaire was structured with a combination of closed-ended and open-ended questions. The closed-ended questions were designed to provide a list or select suitable responses, which were subsequently coded for analysis. On the other hand, the open-ended questions allowed respondents to freely explain their ideas without any constraints. Prior to conducting the interviews, the purpose of the study was clearly explained to the respondents, and their voluntary participation was sought. Moreover, to facilitate effective communication, the questionnaire was translated from English to the local language, Amharic, ensuring that language barriers did not impede the data collection process. Finally, their responses were treated with utmost confidentiality and were not disclosed to anyone beyond the research team.We collected data from September13/2023 to December13/2023.

## 2.3. Sampling technique and sample size determination

In this study, a multi-stage sampling method was employed to select the samples. Firstly, the Raya Kobo District in the North Wollo Zone was purposefully chosen due to its potential for onion production and its significant role in supplying major onion market centers in the Zone. Secondly, five kebeles within the district, namely Adiskigni, Jarota, Kobo Zuria, Aradum, and Abuare were purposively selected based on their high onion production and their connection to major market centers in both the District and the Zone level. In the third stage of the sampling procedure, onion-producing households were selected using simple random sampling technique from each of the selected kebeles for the purpose of conducting interviews. To determine the appropriate sample sizes for the households, the sampling formula developed by [31] was used. It can be expressed as; $n = \frac{N}{1+N(e^2)}$, here, "n" represents the required sample size, "N" denotes the total number of onion-producing households in the target population, and "e" represents the precision level (set at 7% in this study). As a result, the total number of onion-producing households from the five selected kebeles was found to be 2,540. This figure was utilized to determine the appropriate sample size for this study, and as a result, the final sample size of this study was determined to be 189 onion producing farm households ($n = \frac{2540}{1+5(0.07^2)} = 189$). Table 1 provides the sampling distribution across the study kebeles.

**Table 1. Sampling distribution in the study area.**

| Kebeles | Total onion producing households | Number of proportional sampled |
|---------|----------------------------------|-------------------------------|
| Adiskigni | 586 | 44 |
| Jarota | 282 | 21 |
| Kobo Zuria | 643 | 48 |
| Aradum | 684 | 51 |
| Abuare | 345 | 25 |
| **Total** | **2540** | **189** |

Source: District agricultural office, 2022

## 2.4. Method of data analysis

To achieve the study's objective, descriptive statistics was used to provide a comprehensive description of the demographic and socio-economic profiles, farm specific factors as well as the institutional and onion production activities. This involved employing various measures such as frequency, mean, standard deviation, percentages, and charts to present the data effectively. In addition to descriptive statistics, an inferential method of data analysis was employed using the linear production function estimated using Ordinary Least Squares (OLS). This approach facilitated the exploration of significant relationships between demographic factors, socio-economic aspects, institutional factors, farm-specific factors, and the determinants of onion production.

## 2.5. Conceptual framework

Onion is a vital vegetable crop globally, with significant contributions to economy, nutrition, medicinal properties, and flavors. In Ethiopia, it is grown extensively under rain fed in the "Meher" season and under irrigation in the off season for local consumption and exports, playing a crucial role in the rural economy and job creation. Despite its potential for economic and health benefits, production and productivity have not reached the required levels, leading to inconsistent trends and limited benefits for farmers [10, 11, 23] and [29]. Several factors contribute to the low levels of onion production. According to [29] onion production is significantly influenced by gender, age, educational level, production experience, land covered by onion, and household size has a significant influence on onion production; and challenged by timely and adequate supplies, inadequate irrigation and unfair, high cost of major production inputs. Besides, the low productivity could be attributed to the low fertility of soil, inappropriate use of fertilizer, limited availability of quality seeds and minimal applications of modern inputs with appropriate agronomic practices used [11]. Onion production also challenged by high cost of improved varieties, lack of certified accessibility of input, existence of labor which delay adoption, lack of finance and market, lack of awareness and training, poor extension contact and credit services, lack of knowledge and social network [17]. Furthermore, the low onion productivity was found to be attributed to old variety, limited availability of quality seeds, disease and low adoption of recommended production inputs [22].

Accordingly, we hypothesized that large family members, male headed households, educated farmers, obtaining access to irrigation, modern inputs, credit, extension service, and being a member of farm cooperative contribute in the operation of farm activities of onion. Farmers who have an oxen enables them to manage their farm lands, and experienced farmers have better administrative abilities that they can exert to their onion production [11, 14, 15, 17, 19–28] and [29].

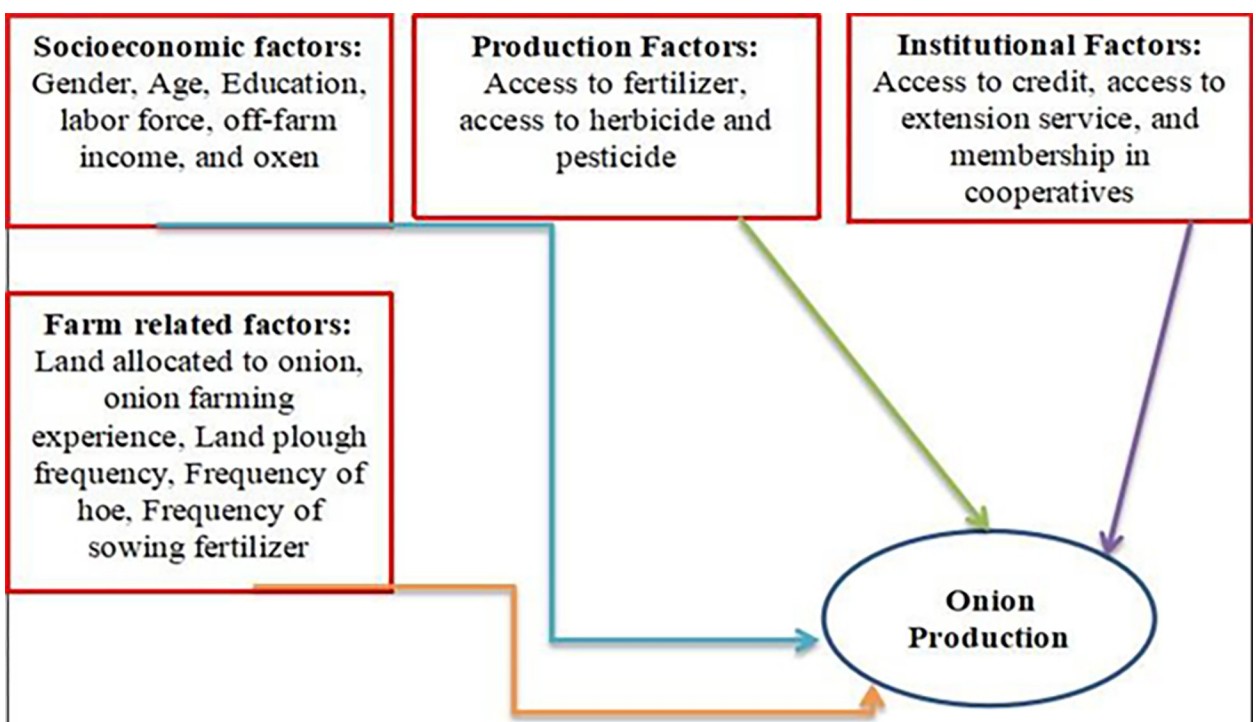

**Fig 2. Conceptual framework of the study, which is adapted from [29].**

Fig 2 presents the conceptual framework of the study, which has been developed based on a thorough review of relevant literature. This framework outlines the relationship among the dependent and a set of explanatory variables. The variables considered in this framework encompass socioeconomic factors, institutional factors, production variables, and farm-specific variables. The socioeconomic variables examined in this study include gender, age, education, labor force, off-farm income, and the number of oxen owned. The production variables taken into account are access to fertilizers, and access to herbicides and pesticides. In terms of the institutional factors, the study considers access to credit, access to extension services, and membership in cooperatives. Lastly, the framework includes important farm-related variables such as the amount of land allocated to onion cultivation, onion farming experience, frequency of land ploughing, hoe usage frequency, and frequency of fertilizer application during plantation.

## 2.6. Model specification and estimation technique

[32] and [33] have highlighted that the most commonly used approach to empirically analyze productivity is to employ a production function as a starting point. A production function, in its broadest sense, establishes a connection between output (Y) and various inputs (X), with both variables potentially being vectors. For empirical analysis, a specific functional form of the production function needs to be selected. In line with [34], it is assumed that a firm's value added is determined by various functional forms, including linear, quadratic, square root, trans-log, and Cobb-Douglas. For the purpose of this research, we have opted to employ the linear production function estimated using the ordinary least squares (OLS) model. This choice was made due to its exceptional flexibility and suitability in analyzing farm-level data when compared to other functional forms. While previous studies [12, 14, 15, 17, 19] and [21]

have predominantly utilized the Cobb-Douglas production function, often in the form of a stochastic production frontier, to estimate the determinants and efficiency of onion production, we have found that the Cobb-Douglas function possesses certain limitations. Specifically, it exhibits reduced flexibility and often fails to yield coefficients with appropriate sign and magnitude. Additionally, when estimating a trans-log production function, multicollinearity among the explanatory variables tends to be present due to the degrees of freedom associated with the model.

Moreover, it is worth noting that the aforementioned models assume that all explanatory variables must be continuous. However, in the context of onion production, there exist both continuous and categorical variables that influence the outcome. Consequently, the chosen linear production function model holds great potential, as it allows for the incorporation of explanatory variables in various forms, thereby enabling the generation of efficient estimates [25]. Accordingly, the estimation of the production function focuses on the linear production function model. It is defined as follows:

$$Y_i = \alpha_0 + \beta_i X_i + \varepsilon_i \qquad 1$$

Eq (1) represents the basic linear regression model that we seek to estimate, where $Y_i$ is a vector of onion production; $X_i$'s are a set of explanatory variables which affects onion production including demographic, socioeconomic, production, institutional and farm related factors; $\alpha_0$ and $\beta_i$'s are parameters to be estimated, and $\varepsilon_i$ is a vector of error term, which is assumed to follow an independent-normal distribution with a mean of zero and a constant variance of $\sigma 2$.

## 2.7. Ethical considerations

Ethical approval was obtained for this study from the Research Ethics Approval Committee of Woldia University, Ethiopia, and authorized by the Institutional Review Board (IRB) committee of faculty of Business and Economics of Woldia University under Ref. No: FBE/RCSTT/216/2023. All participants provided written informed consent and authorizations to request the necessary data for analyzing the determinants of onion production practices in the area. The analysis was conducted using data that does not contain any information that could lead to the identification of the participants.

# 3. Result and discussions

## 3.1. Descriptive summary

A variable description, measurement, a descriptive summary of the variables of onion production practice and hypothesis of the study variables is shown in Table 2. The table reveals that male-headed households constitute the majority of onion producers (87%), while female-headed households account for a smaller proportion (13%). Among the total respondents, 39% were found to be illiterate, while approximately 44%, 10%, and 7% had completed educational levels ranging from grades 1–8, grades 9–12, and above grade 12 (Diploma and Degree), respectively.

Regarding off-farm activities, only 17% of the respondents were actively engaged, while the majority (83%) did not participate in such activities, indicating that a significant proportion of onion producers in the study area did not engage in off-farm work. In terms of credit access, only 24% of the respondents had access to credit, while the remaining 76% did not. Furthermore, approximately 59% of the respondents had received extension services for their onion production practices, and also 36% were members of farm cooperatives in the study area.

**Table 2. Description and summary of variables used for the regressions.**

| Variables | Description and measurement | Expected sign | Category | Frequency |
|---|---|---|---|---|
| Onion production | Onion output per hectare, we used log of onion production to produce consistent estimates of the parameter | | | |
| Gender | Sex of the household head = 1 if male; 0 if female | +/- | Male | 164(87) |
| | | | Female | 25(13) |
| Education | Household Education level = 0 if illiterate; 1 if grade 1–8; 2 if grade 9–12; 3 if above grade 12 | + | Illiterate | 74(39) |
| | | | Grade 1–8 | 83(44) |
| | | | Grade 9–12 | 19(10) |
| | | | Above 12 | 13(7) |
| Off-farm Employment | = 1 if the farm household had participated; 0 otherwise | +/- | Yes | 32(17) |
| | | | No | 157(83) |
| Access to credit | = 1 if the farm household had taken loan; 0 otherwise | + | Yes | 45(24) |
| | | | No | 144(76) |
| Extension visit | = 1 if household had extension visit during their practice; 0 otherwise | + | Yes | 111(59) |
| | | | No | 78(41) |
| Farm cooperative | = 1 if the household is member of farm cooperative;0 otherwise | + | Yes | 68(36) |
| | | | No | 121(64) |
| Irrigation access | = 1 if the household had access to irrigation; 0 otherwise | + | Yes | 172(91) |
| | | | No | 17(9) |
| Fertilizer | = 1 if the farm household had access to fertilizer; 0 otherwise | + | Yes | 179(95) |
| | | | No | 10(5) |
| Pesticide | = 1 if the farm household had access to pesticide; 0 otherwise | + | Yes | 145(77) |
| | | | No | 44(23) |
| Herbicide | = 1 if the farm household had access to herbicide; 0 otherwise | + | Yes | 119(63) |
| | | | No | 70(37) |
| (Percent's in parenthesis) | | | | |
| | | | **Mean** | **Std. dev.** |
| Age | Age of the household head, measured in years, we have taken log of Age to produce consistent estimators | +/- | 51.89 | 12.84 |
| Labour force | The total number of labour force of the household | + | 2.64 | 0.88 |
| Experience | Farm household experience in onion production in years | + | 4.51 | 2.38 |
| Farm size | Total farm size of the household used for onion in hectare | + | 0.64 | 0.23 |
| Oxen | Total oxen size of the household measured in number | + | 1.63 | 1.10 |
| Sowing of fertilizer | Frequency of employment of chemical fertilizer on the onion plant after transplanting | - | 1.77 | 0.58 |
| Land plough | Frequency of the onion site ploughed before transplantation | + | 3.66 | 0.66 |
| Farm hoe | Frequency of hoeing the onion field after transplantation | - | 1.72 | 0.49 |
| Onion output/ha | Onion output per hectare | | 83.18 | 49.53 |

Source: own survey, 2023

These findings suggest that a considerable number of farm households in the study area did not have access to extension services and were not actively involved in farm cooperatives.

Regarding input availability, the majority of the farm households (91%, 95%, 77%, and 63%) had access to irrigation water, fertilizer inputs, pesticides, and herbicides, respectively, for their onion cultivation. The average age of the farm household head involved in onion production was found to be 51.89 years, and the average number of laborers per farm household was 2.64. On average, the farm households had 4.51 years of experience in onion production,

and the average farm size allocated for onion cultivation was 0.64 hectares. Additionally, the average number of oxen owned by the farm households was 1.63.

The table also presents the mean frequency of land plowing, frequency of fertilizer application after transplanting, and frequency of hoeing the onion field after transplanting, which were reported as 3.66 times, 1.77 times, and 1.72 times, respectively. Finally, the average onion production size per hectare by the farm households in the study area was 83.18 quintals, which is lower than the average reported for the Amhara National Regional State (110.67 quintals/ha) by the [35]. This disparity can be attributed to various production constraints faced by onion producers in the study area, ranging from challenges related to input access to issues surrounding harvesting practices.

## 3.2. Econometric analysis

The OLS estimates are presented in Table 3, providing insights into the factors influencing onion production in the study area. The data were checked for possible outliers before running for statistical analysis. Preliminary descriptive analysis was carried out to identify inconsistencies and irregularities in data entry, which were then corrected by crosschecking the questionnaires. Prior to the estimation, various diagnostic tests were conducted to ensure the data reliability and model's suitability. The results of these tests confirmed that the model fits the data well, with a rejection of the joint equality of regression coefficients (Prob>F = 0.000); the

**Table 3. Multiple linear regression model estimates.**

| Variables | Coefficient | Std. err |
|---|---|---|
| Gender | .336 | .097*** |
| Education (base: illiterate) | | |
| Grade 1–8 | .026 | .132 |
| Grade 9–12 | .120 | .084 |
| Grade >12 | .143 | .017*** |
| Ln Age | -.013 | .052 |
| Experience | .118 | .028*** |
| Labour force | .771 | .154*** |
| Farm size | .093 | .055* |
| Oxen | .033 | .108 |
| Off-farm Employment | -.181 | .115 |
| Access to credit | -.084 | .089 |
| Extension visit | .334 | .111*** |
| Farm cooperative | .129 | .263 |
| Irrigation access | .542 | .088*** |
| Fertilizer | .929 | .166*** |
| Herbicide | -.109 | .096 |
| Pesticide | .146 | .104 |
| Land plough frequency | .663 | .184*** |
| Frequency of hoe | -.084 | .092 |
| Frequency of sowing fertilizer | -.209 | .102** |
| Constant | 2.86 | 0.34*** |

Source: Own computation, 2023

*, ** and *** signifies level of significance at 10%, 5% and 1% respectively.

Note: Obs. = 189; $F_{(20, 168)}$ = 26.95, Prob > F = 0.0000*; $R^2$ = 0.7880.

data has no serious multicollinearity problem (VIF = 1.23), and the data has no heteroskedasticity problem (Prob>Chi$^2$ = 0.2607). The findings indicate that the gender of the household head has a positive and significant influence on onion production among farm households. Specifically, a 1% increase in male-headed onion producer's results in a 33.6% increase in production, all other factors held constant. This suggests that male-headed households are the primary contributors to onion production in the study area. Conversely, female-headed farmers are discouraged from producing onions due to various factors such as time constraints, limited working capital, and difficulties in selling onions and other high-value crops in Ethiopia [29].

Furthermore, the education level of the household head significantly affects onion production in the study area. Compared to illiterate households, an increase in the number of farm households with education beyond grade 12 leads to a 14.3% increase in onion production, assuming all other variables remain constant. This implies that households with higher levels of education are more likely to engage in onion production, as education enhances their knowledge, skills, and access to information, consequently boosting their production and supply to the market [25].

Moreover, farming experience in onion production plays a positive and significant role in determining the success rate of farmers in producing onions. An increase of one year in farming experience results in an 11.8% increase in the success rate of onion production, assuming other factors remain unchanged. This indicates that experience in years that the household would have engaged in onion production improves the household's abilities in production as farmers becoming more familiar with the production environment and gaining greater expertise in onion cultivation; and also experienced farmers have better administrative abilities that they can exert to their production and handling [11].

The number of laborers in farm households also has a positive and significant impact on onion production, with an increase of one laborer leading to a 77.1% increase in production, assuming all else remains constant. This suggests that households with a larger labor force are more likely to engage in onion production in the study area. This finding aligns with previous research that highlights the common practice in developing countries where farmers rely on family labor. Having an adequate and active labor force enables timely and efficient execution of production processes, thereby improving agricultural output; again The large family members contribute significantly to the operation of the farm's onion cultivation activities, providing an additional and abundant source of labor [11].

The size of the cultivated onion farm also significantly influences onion production, with an increase of one unit in the farm size allocated for onion production resulting in a 9.3% increase in production, assuming other variables are held constant. This confirms that farmers with large larger farm sizes allow farmers to produce a greater quantity of onions, thereby increasing the supply available in the market and enabling farmers to derive greater benefits from onion production [25].

The variable extension visit positively and significantly affects onion production in the study area. As the frequency of extension visits to farm household's increases, onion production improves by 33.4%, assuming all other factors remain constant. This is because farmers who have frequent contact with extension agents gain access to valuable information on production techniques and can adopt better technologies, resulting in increased onion production for the market [11, 36].

Furthermore, access to irrigation water has a positive and significant effect on onion production, with an increase in the number of farm households having access to irrigation water resulting in a 54.2% increase in production, assuming all other variables remain constant. This indicates irrigation access can provide water to onion production when natural rainfall is insufficient or unreliable, ensuring their optimal growth, development, and productivity; and

can cultivate onions multiple times per year, leading to higher onion supplies in the market [25].

The availability of fertilizer also has a positive and significant impact on onion production. A 1% increase in the number of farm households with access to fertilizer leads to a 92.9% increase in production, assuming other factors are held constant. This indicates that the use of fertilizer enhances onion production by improving soil fertility and providing essential nutrients to the onion crop. Moreover, the application of fertilizers improve onion production by supplying essential nutrients that are necessary for the plant's growth and development. These nutrients, such as nitrogen, phosphorus, and potassium, promote root development, enhance bulb formation, and increase overall yield [11, 37].

Moreover, the frequency of land plowing for onion production positively and significantly determines onion production. An increase of one time in the frequency of land plowing results in a 66.3% increase in onion production, assuming all other variables remain constant. This is likely because onion cultivation is challenging in heavy and compacted soils, necessitating frequent tillage to create a favorable environment for onion growth and combat harmful microorganisms. This finding is consistent with the work of [16].

On the other hand, the frequency of sowing fertilizer after transplantation negatively and significantly affects onion production. An increase in the frequency of sowing fertilizer leads to a 20.9% decrease in onion production, assuming all other factors are held constant. This implicates that excessive and frequent application of chemical fertilizer stimulates leaf growth more than bulb growth, making the plants susceptible to diseases such as rust and fungus [29]. Lastly, some explanatory variables were found to have an insignificant effect on onion production. However, these insignificant variables are consistent with the theoretical framework and hypotheses of the study, suggesting that they do not play a significant role in explaining the variations in onion production in the study area.

In summary, the results reveal that the gender of the household head, education level, farming experience, labor force size, farm size, extension visits, access to irrigation water, availability of fertilizer, frequency of land plowing, and frequency of sowing fertilizer after transplantation all have significant impacts on onion production among farm households in the study area.

### 3.3. Farm size distribution for onion production in the study area

To enhance the robustness of our findings, we have included qualitative data and analysis for some variables. Fig 3 illustrates the distribution of respondents based on their allocation of farm land for onion production in the study area. The results indicate that approximately 43%, 23%, 19%, and 6% of farm households have allocated 0.25 ha, 0.5 ha, 0.75 ha, and 1 ha, respectively, for onion production. Additionally, a small proportion of respondents (around 4%, 3%, and 2% of the total) have allocated 1.25 ha, 1.5 ha, and 2 ha or more for onion cultivation. This suggests that only a few farm households engage in onion production on a larger scale. Consequently, the overall level of onion production and investment in the district is minimal, resulting in limited output that mainly supply to local markets only.

### 3.4. Major challenges of onion production in the study area

Onion-growing farmers were asked a specific set of questions to identify the challenges they face during onion cultivation. They were provided with closed-ended questions that addressed the constraints they encounter. Based on their responses, farmers identified several major constraints, as depicted in Fig 4. These constraints include shortages of fertilizer and other inputs, high input costs, crop diseases, labor costs, soil infertility, rain during harvesting, simultaneous

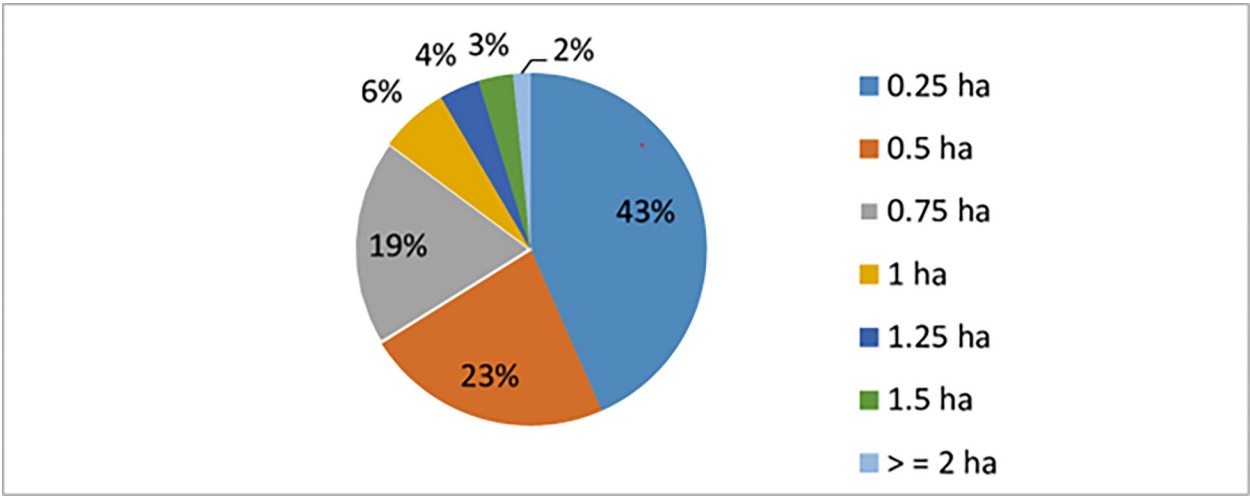

**Fig 3. Farm size for onion production.** *Source: Own survey, 2023.*

planting periods, land shortages, lack of awareness regarding storage methods, and improper use of inputs. These findings align with the research conducted by [29] and [38], which also highlighted challenges such as high post-harvest losses, difficulties in market linkage, low market prices, limited knowledge on extending the shelf life of onions (resulting in rotting and sprouting), absence of suitable long-lasting onion varieties, low prices during peak harvesting

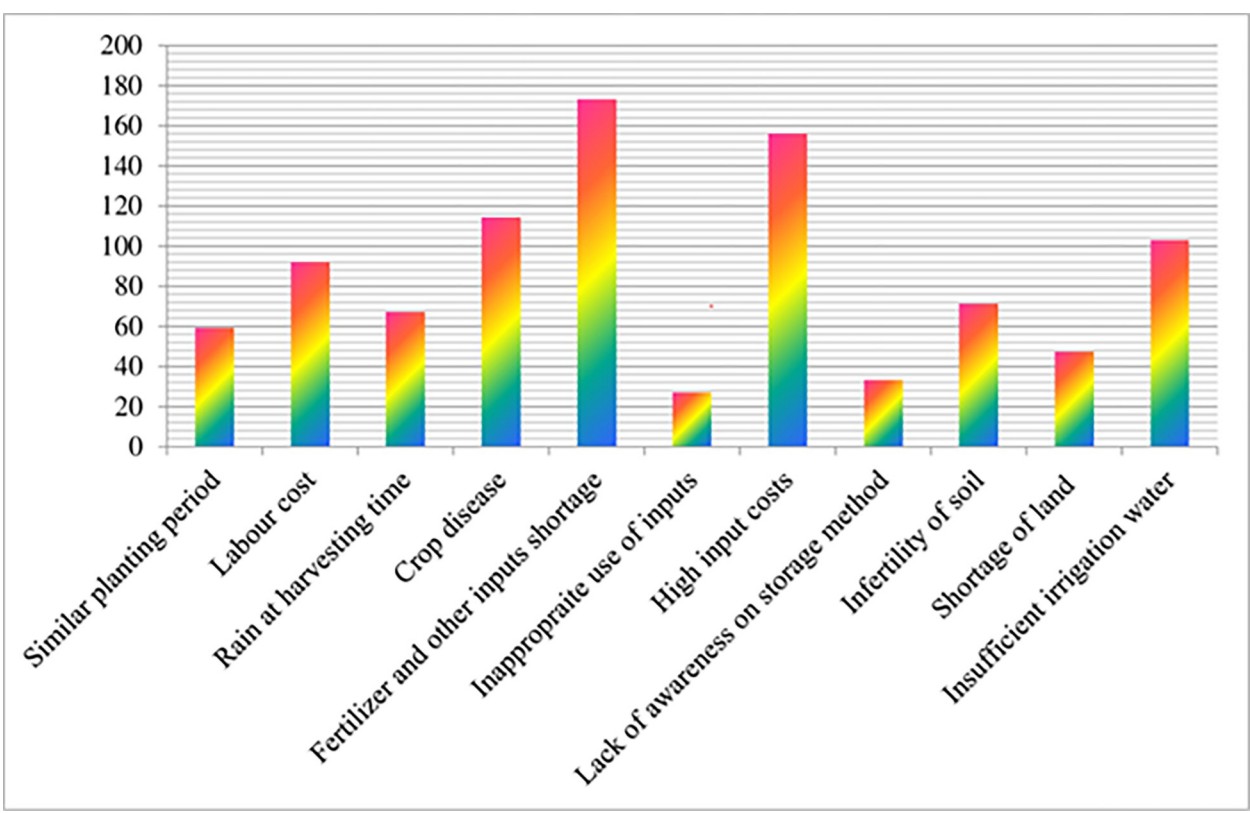

**Fig 4. Major onion production constraints in the study area.** *Source: Own survey, 2023.*

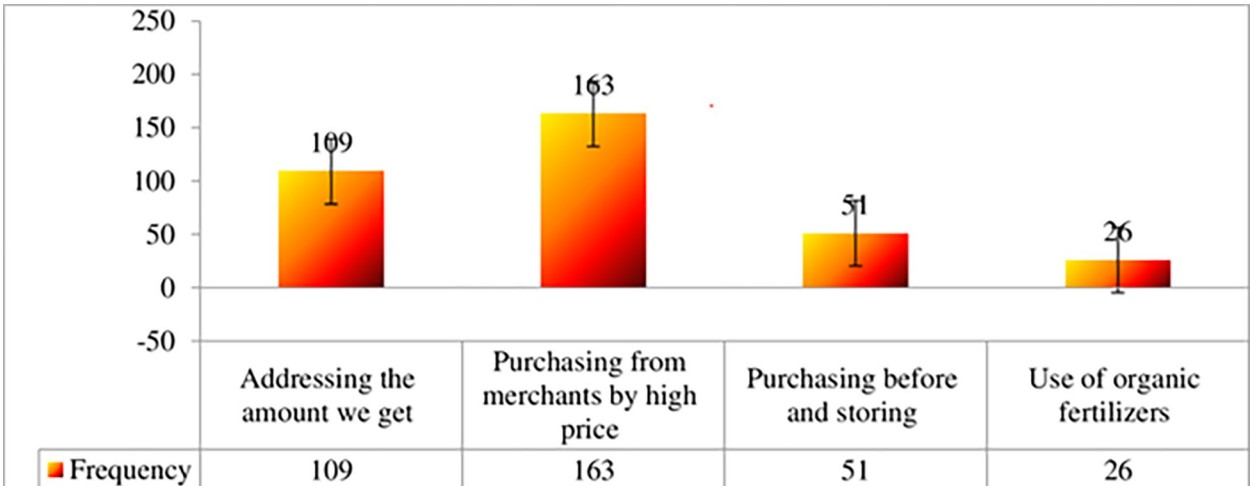

**Fig 5. Alternatives to solve the problem of fertilizer and other inputs shortage.** *Source*: *Own survey*, *2023*.

time, limited access to fertilizers and high prices, absence of storage facilities, poor road access, diseases and insect pests, lack of awareness regarding storage methods, interference of brokers in market information, rain during the maturity period, water scarcity for irrigation, lack of quality seeds, fuel shortages, high labor costs, limited extension support, and simultaneous planting schedules.

Fig 5 presents the various approaches adopted by onion producers in the study area to address the issue of fertilizer and other input shortages during their onion production practices. Among the surveyed farm households, the majority have employed several strategies to tackle this challenge. These strategies include purchasing the required inputs from merchants at a higher cost, procuring them based on the available quantity, buying in advance and storing them until the production period, and incorporating organic fertilizers alongside chemical fertilizers like Urea and Dap.

Based on the descriptive summary, approximately 41% of the farm households in the study area did not receive extension services. Fig 6 illustrates the reasons provided by these

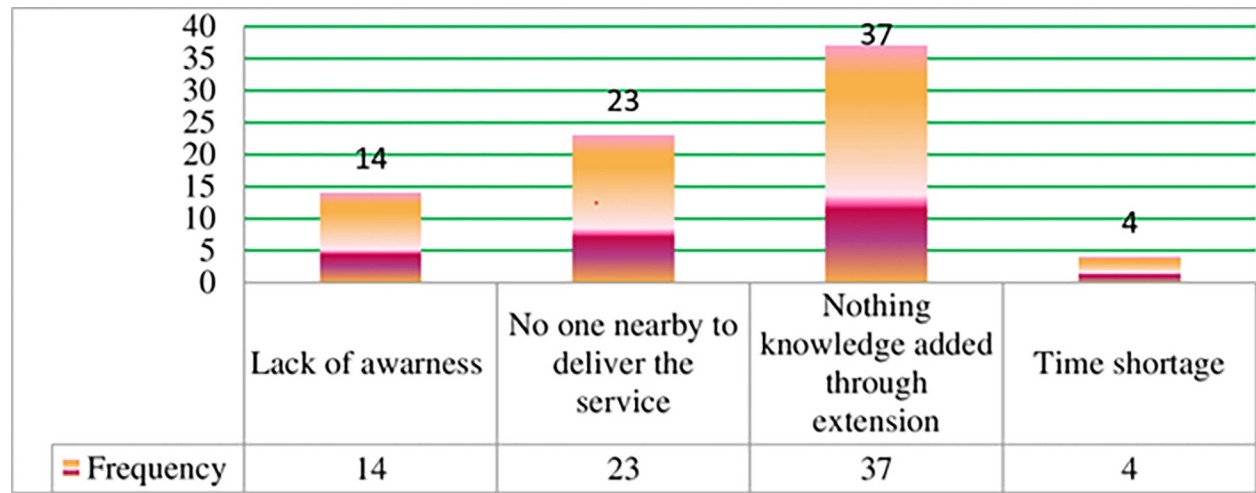

**Fig 6. The reason for didn't get extension visit during their practice.** *Source*: *Own survey*, *2023*.

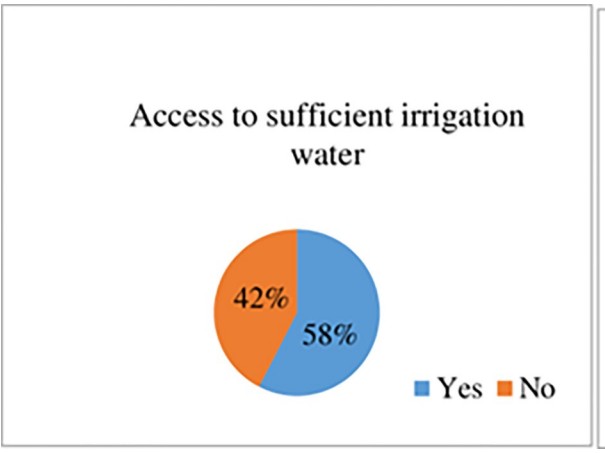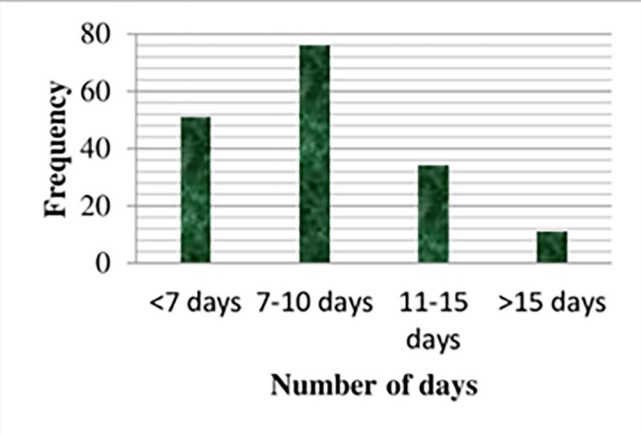

**Fig 7. Irrigation practices in the study area.** *Source*: *Own survey, 2023.*

households for not accessing extension services during their onion production practices. The findings indicate that the majority of respondents agreed that extension services offered limited knowledge and were considered a waste of their valuable working time. They believed that extension workers had limited expertise, and farmers themselves possessed ample knowledge and experience in onion cultivation. Other reasons cited included the absence of timely service providers, lack of awareness about the availability of extension services, and insufficient time to access the services even if they were available. Furthermore, it is worth noting that the district has experienced conflicts and other events over 5 years, resulting in a scarcity of extension workers to provide convenient and timely services.

Finally, Fig 7 depicts the irrigation practices in the study area. Consistent with the descriptive summary, onion production primarily relied on irrigation methods. The majority of onion producers (91%) utilized irrigation for their onion cultivation. However, the accessibility to irrigation water varied among farmers. Among the total respondents, approximately 58% had sufficient access to irrigation water and applied it at intervals of 7 to 10 days or less. Conversely, around 42% of the respondents faced inadequate access to irrigation water, resulting in the application of irrigation water every 11 to 15 days or more than 15 days.

## 4. Conclusions and recommendations

In conclusion, this study examined the factors influencing onion production in the Raya Kobo district of Amhara regional state of Ethiopia. To attain this objective, data was gathered from 189 onion-producing farm households through structured questionnaire, and analyzed using descriptive and econometric techniques. The results of the study indicated that gender of the household head, age of the household head, educational level of the household head, labour force size of the household, onion farming experience, onion farm size, access to extension services, irrigation water, fertilizer availability, frequency of land plough, frequency of hoe and frequency of sowing fertilizer after transplantation have a significant effect on onion production. In addition, onion producing farmers faced constraints such as fertilizer and input shortages, high input costs, crop diseases, labor costs, soil infertility, and inadequate knowledge of storage methods. These findings highlight the importance of considering various demographic, socio-economic institutional, production and farm specific factors in promoting and improving onion production for farmers. Thus, we recommend that policymakers and stakeholders can utilize these findings to develop appropriate policies and interventions to enhance

onion production in the district include promoting gender equality, implementing education and training programs, strengthening extension services, developing irrigation infrastructure, and improving access to inputs. Furthermore, we suggest that the concerned body should lay effective awareness creation programs to train the growers about appropriate usage of farm technologies and the farmers to plough their onion land more frequently before transplantation. These measures aim to empower farmers, enhance their knowledge and skills, ensure water availability, and facilitate access to necessary resources. Thus, minimizing the onion farmers' constraints can contribute to increased onion production, and farmers' well-being may improve significantly.

## Author Contributions

**Conceptualization:** Mesele Belay Zegeye, Tesfahun Ayanaw Alemu, Moges Asmare Sisay, Sisay Genanu Mulaw, Tadesse Wudu Abate.

**Data curation:** Mesele Belay Zegeye, Moges Asmare Sisay, Tadesse Wudu Abate.

**Formal analysis:** Mesele Belay Zegeye, Tesfahun Ayanaw Alemu, Moges Asmare Sisay, Sisay Genanu Mulaw, Tadesse Wudu Abate.

**Investigation:** Mesele Belay Zegeye.

**Methodology:** Mesele Belay Zegeye, Tadesse Wudu Abate.

**Software:** Mesele Belay Zegeye.

**Writing – original draft:** Mesele Belay Zegeye.

**Writing – review & editing:** Mesele Belay Zegeye, Tesfahun Ayanaw Alemu, Moges Asmare Sisay, Sisay Genanu Mulaw, Tadesse Wudu Abate.

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
