## [Decision Letter · Decision Letter 0]

18 Apr 2024

PONE-D-24-08353Factors Affecting Onion Production: an Empirical Study in the Raya Kobo District, Amhara Regional State, EthiopiaPLOS ONE

Dear Dr. Zegeye,

Thank you for submitting your manuscript to PLOS ONE. After careful consideration, we feel that it has merit but does not fully meet PLOS ONE’s publication criteria as it currently stands. Therefore, we invite you to submit a revised version of the manuscript that addresses the points raised during the review process.

We look forward to receiving your revised manuscript.

Kind regards,

Sumit Jangra, Ph.D.

Academic Editor

PLOS ONE

2. In the online submission form, you indicated that [Data will be made available on request.].

4. Please ensure that you include a title page within your main document. You should list all authors and all affiliations as per our author instructions and clearly indicate the corresponding author.

Reviewers' comments:

Reviewer's Responses to Questions

**Comments to the Author**

1. Is the manuscript technically sound, and do the data support the conclusions?

Reviewer #1: Yes

Reviewer #2: Partly

2. Has the statistical analysis been performed appropriately and rigorously? 

Reviewer #1: Yes

Reviewer #2: Yes

3. Have the authors made all data underlying the findings in their manuscript fully available?

Reviewer #1: Yes

Reviewer #2: Yes

4. Is the manuscript presented in an intelligible fashion and written in standard English?

Reviewer #1: Yes

Reviewer #2: Yes

5. Review Comments to the Author

Reviewer #1: Factors Affecting Onion Production: an Empirical Study in the Raya Kobo District,

Amhara Regional State, Ethiopia

Reviewer comments:

The overall structure of the paper needs improvement. The abstract part is shallow and lacks the research gap and others. The introduction does not provide a clear roadmap for the study, and the flow of ideas throughout the paper is not well organized.

The methodology section lacks sufficient detail on the data collection process, sample size, and sampling techniques. It's important to provide a clear understanding of how the data was gathered and analyzed.

The choice of statistical methods should be justified more thoroughly. Were there any alternative methods considered, and if so, why were they not chosen?

The paper lacks a discussion on data quality and potential biases in the dataset. It's crucial to assess the reliability of the data and discuss any limitations that might affect the study's findings.

The theoretical literature review is essential to provide a more comprehensive overview of the existing literature, including recent developments in the field.

The paper introduces a theoretical framework, but it's not well-integrated with the research questions or hypotheses. The authors should clarify how the theoretical framework guides the study.

The results are presented in a somewhat fragmented manner. The authors should consider reorganizing them to make the findings more coherent and easier to follow.

The discussion of results is relatively brief and lacks critical analysis. The authors should investigate deeper into the implications of their findings and relate them to the existing literature.

The conclusion is somewhat abrupt and doesn't provide a robust summary of the paper's main contributions and their implications for policy and practice. A more detailed discussion of policy recommendations is needed.

The paper contains several grammatical and stylistic errors. The authors should carefully proofread and edit the manuscript for clarity and consistency.

These comments should help the authors improve their paper by addressing key aspects of clarity, abstract, introduction, methodology, literature review, theoretical framework, results, discussion, recommendation, and overall presentation.

Reviewer #2: Coherence is lacking among the structured Abstract, Result and discussion, Conclusion and recommendations. Likewise, the author mentioned supply chain in abstract, while value chain and sustainability in conclusion and recommendations. These are different three domains, While the study investigated only factors affecting onion production according to the drawn conceptual framework in the Figure 1.

Secondly, author mentioned onion farmers' constraints including pot-harvest handling in section 3.1 paragrapg#3 of results & discussions. While the same problem is not discussed in section 3.4 entitled major constraints of onion farmers in study area.

Thirdly, study defined about post-harvest losses on section 4. it's not previously discussed anywhere.

Fourthly, according to the defined study, by minimizing the onion farmers' constraints, farmers' well-being may improve, crop yield may increase that may impact sectoral growth. But will not stronger the value chain or its' sustainability as defined in the last paragraph.

6. PLOS authors have the option to publish the peer review history of their article (what does this mean?). If published, this will include your full peer review and any attached files.

Reviewer #1: No

Reviewer #2: **Yes: **Azhar Rasool

---

## [Author Response · Author response to Decision Letter 0]

2 May 2024

Attached under 'Response to Reviewers' section.

---

## [Decision Letter · Decision Letter 1]

17 May 2024

PONE-D-24-08353R1Factors affecting onion production: an empirical study in the Raya Kobo district, Amhara regional state, EthiopiaPLOS ONE

Dear Dr. Zegeye,

Thank you for submitting your manuscript to PLOS ONE. After careful consideration, we feel that it has merit but does not fully meet PLOS ONE’s publication criteria as it currently stands. Therefore, we invite you to submit a revised version of the manuscript that addresses the points raised during the review process.

We look forward to receiving your revised manuscript.

Kind regards,

Sumit Jangra, Ph.D.

Academic Editor

PLOS ONE

Journal Requirements:

**Additional Editor Comments:**

Accepted with minor revision

Reviewers' comments:

Reviewer's Responses to Questions

**Comments to the Author**

1. If the authors have adequately addressed your comments raised in a previous round of review and you feel that this manuscript is now acceptable for publication, you may indicate that here to bypass the “Comments to the Author” section, enter your conflict of interest statement in the “Confidential to Editor” section, and submit your "Accept" recommendation.

Reviewer #1: All comments have been addressed

Reviewer #2: All comments have been addressed

Reviewer #3: All comments have been addressed

2. Is the manuscript technically sound, and do the data support the conclusions?

Reviewer #1: Yes

Reviewer #2: Yes

Reviewer #3: Yes

3. Has the statistical analysis been performed appropriately and rigorously? 

Reviewer #1: Yes

Reviewer #2: Yes

Reviewer #3: Yes

4. Have the authors made all data underlying the findings in their manuscript fully available?

Reviewer #1: Yes

Reviewer #2: Yes

Reviewer #3: Yes

5. Is the manuscript presented in an intelligible fashion and written in standard English?

Reviewer #1: Yes

Reviewer #2: Yes

Reviewer #3: Yes

6. Review Comments to the Author

Reviewer #1: Factors affecting onion production: an empirical study in the Raya Kobo district, Amhara regional state, Ethiopia

The authors address most of the issues raised. But, still it needs some modifications (like abstract part, the study area, and others have been given comments on manuscripts).

Reviewer #2: The authors addressed each comment/observation and the attached the revised manuscript have been accepted for further proceedings.

Reviewer #3: 1.On the introduction section, if The Amhara Region contributes around 50% of the national onion production, why onion production is low compared to other region?

2.If Raya Kobo district is recognized as a major onion producer in Amhara region, it is better to support this sentence with empirical evidence.

3.The types of onion seeds are one of the determinants of onion production by itself, how do you see this?

4.Add the map of the study area.

7. PLOS authors have the option to publish the peer review history of their article (what does this mean?). If published, this will include your full peer review and any attached files.

Reviewer #1: No

Reviewer #2: No

Reviewer #3: No

---

## [Author Response · Author response to Decision Letter 1]

22 May 2024

Journal: PLOS ONE

Manuscript No: PONE-D-24-08353

Title: Factors Affecting Onion Production: an Empirical Study in the Raya Kobo District, Amhara Regional State, Ethiopia

The authors are thankful to reviewers for their comments to improve the quality of our manuscript.

Please find below our response to the each of the comments in the order in which they have been raised.

Reviewer 1: 

Reviewer #1: The authors address most of the issues raised. But, still it needs some modifications (like abstract part, the study area, and others have been given comments on manuscripts).

Response: Thank you for your encouraging comments. We have checked the abstract, study area and others, and revised it accordingly.

Reviewer 2: 

The authors addressed each comment/observation and the attached the revised manuscript have been accepted for further proceedings.

Response: Thank you for your appreciation. 

Reviewer 3: 

Reviewer #3: 

On the introduction section, if The Amhara Region contributes around 50% of the national onion production, why onion production is low compared to other region?

Response: Thank you for your comments. We have revised and corrected according to your suggestion in this version- see section 1.

If Raya Kobo district is recognized as a major onion producer in Amhara region, it is better to support this sentence with empirical evidence.

Response: Thank you for your comments. We have revised and corrected according to your suggestion in this version- see section 3.1.

The types of onion seeds are one of the determinants of onion production by itself, how do you see this?

Response: Thank you for your comments. The types of onion seeds are an important determinant of onion production. In the study area, we found that only one type of onion seed, the red bulb variety, is commonly used by farmers. As a result, we decided to exclude the variable of onion seed type from our analysis, since there was effectively no variation in this factor across the sample.

Add the map of the study area.

Response: Thank you for your comments. We have added it.

---

## [Editor Report · Decision Letter 2]

24 May 2024

Factors affecting onion production: an empirical study in the Raya Kobo district, Amhara regional state, Ethiopia

PONE-D-24-08353R2

Dear Dr. Zegeye,

We’re pleased to inform you that your manuscript has been judged scientifically suitable for publication and will be formally accepted for publication once it meets all outstanding technical requirements.

Kind regards,

Sumit Jangra, Ph.D.

Academic Editor

PLOS ONE
---

## [Editor Report · Acceptance letter]

5 Jun 2024

PONE-D-24-08353R2 

PLOS ONE

Dear Dr. Zegeye, 

I'm pleased to inform you that your manuscript has been deemed suitable for publication in PLOS ONE. Congratulations! Your manuscript is now being handed over to our production team.

Kind regards, 

on behalf of

Dr. Sumit Jangra 

Academic Editor

PLOS ONE